# Integrated Metabolomics and Lipidomics Reveal High Accumulation of Glycerophospholipids in Human Astrocytes under the Lipotoxic Effect of Palmitic Acid and Tibolone Protection

**DOI:** 10.3390/ijms23052474

**Published:** 2022-02-23

**Authors:** Ricardo Cabezas, Cynthia Martin-Jiménez, Martha Zuluaga, Andrés Pinzón, George E. Barreto, Janneth González

**Affiliations:** 1Grupo de Investigación en Ciencias Biomédicas GRINCIBIO, Facultad de Medicina, Universidad Antonio Nariño, Bogota 110231, Colombia; 2Division of Neuropharmacology and Neurologic Diseases, Yerkes National Primate Research Center, Atlanta, GA 30301, USA; cmart51@emory.edu; 3Escuela de Ciencias Básicas Tecnologías e Ingenierías, Universidad Nacional Abierta y a Distancia, Bogota 111511, Colombia; viviana.zuluagarojas@gmail.com; 4Grupo de Investigación en Cromatografía y Técnicas Afines, Universidad de Caldas, Manizales 170002, Colombia; 5Laboratorio de Bioinformática y Biología de Sistemas, Universidad Nacional de Colombia-Bogotá, Bogota 111321, Colombia; ampinzonv@unal.edu.co; 6Department of Biological Sciences, University of Limerick, V94 T9PX Limerick, Ireland; george.barreto@ul.ie; 7Health Research Institute, University of Limerick, V94 T9PX Limerick, Ireland; 8Departamento de Nutrición y Bioquímica, Facultad de Ciencias, Pontificia Universidad Javeriana Bogotá, Bogota 110231, Colombia

**Keywords:** lipidomics, tibolone, palmitic acid, astrocytes

## Abstract

Lipotoxicity is a metabolic condition resulting from the accumulation of free fatty acids in non-adipose tissues which involves a series of pathological responses triggered after chronic exposure to high levels of fatty acids, severely detrimental to cellular homeostasis and viability. In brain, lipotoxicity affects both neurons and other cell types, notably astrocytes, leading to neurodegenerative processes, such as Alzheimer (AD) and Parkinson diseases (PD). In this study, we performed for the first time, a whole lipidomic characterization of Normal Human Astrocytes cultures exposed to toxic concentrations of palmitic acid and the protective compound tibolone, to establish and identify the set of potential metabolites that are modulated under these experimental treatments. The study covered 3843 features involved in the exo- and endo-metabolome extracts obtained from astrocytes with the mentioned treatments. Through multivariate statistical analysis such as PCA (principal component analysis), partial least squares (PLS-DA), clustering analysis, and machine learning enrichment analysis, it was possible to determine the specific metabolites that were affected by palmitic acid insult, such as phosphoethanolamines, phosphoserines phosphocholines and glycerophosphocholines, with their respective metabolic pathways impact. Moreover, our results suggest the importance of tibolone in the generation of neuroprotective metabolites by astrocytes and may be relevant to the development of neurodegenerative processes.

## 1. Introduction

Lipotoxicity is a metabolic condition resulting from the accumulation of free fatty acids in non-adipose tissues [1,2]. This condition involves a series of pathological responses triggered after chronic exposure to high levels of fatty acids, which may be detrimental to cellular homeostasis and viability, leading to cellular dysfunction, lipid droplet formation, and cell death [3,4]. Although the exact mechanisms related to a lipotoxic event have not been thoroughly characterized, this process is a well-known factor present in obesity and neurodegenerative diseases (NDs) [5,6,7]. In these aspects, lipotoxicity may affect both neurons and astrocytes in the central nervous system (CNS). Astrocytes play a critical role in the CNS homeostasis [8,9], including maintenance of brain metabolism, the promotion of neurovascular coupling, the attraction of cells through the release of chemokines, K^+^ buffering, release of gliotransmitters and glutamate through calcium signaling, control of brain pH, metabolization of dopamine and other substrates by monoamine oxidases, uptake of glutamate and γ-aminobutyricacid (GABA) by specific transporters [10], antioxidative and metabolic support to neurons [11,12], as well as triggering neuroprotection through the activation of various survival signalling cascades and sincitial networks [13,14]. Importantly, lipid metabolism is crucial to the normal astrocytic function and in neuron-astrocyte crosstalk during processes such membrane fluidity, cell signalling, inflammation, and energy generation. During pathological processes such as obesity, ischemia/reoxygenation, multiple sclerosis and Parkinson and Alzheimer diseases, the normal homeostatic functions of the astrocytes are significantly impaired, which can lead to neurodegeneration and related processes [15,16,17].

Both in vitro and in vivo studies suggest that palmitic acid (PA) stimulation on astrocytes induces changes in the expression of mitochondrial genes, pro-inflammatory cytokines, oxidative stress, and morphological changes [18,19,20]. Interestingly, several studies have shown that the steroid compound tibolone exerts beneficial effects in acute and chronic NDs [21,22,23] both in neurons and astrocytes. Likewise, previous studies have shown that tibolone pre-treatment induces antioxidative protection and mitochondrial protection against PA damage in astrocytes [22,24].

This study presents a lipidomic profile of the exo- and endo-metabolome of astrocytic cells, which were generated by means of different treatments: (i) Tibolone; (ii) PA and (iii) PA + tibolone. The measurement of the 3843 metabolites was conducted using LC-QTOF MS/MS and the data processing was performed using MS-DIAL 4.0. Data annotation was carried out by manual comparison of MS/MS spectra and accurate masses of the precursor ion to spectra given in the Fiehn Laboratory’s LipidBlast spectra library [25]. Upon normalization of the abundance values retrieved, several analyses were conducted, including a principal component analysis (PCA), partial least squares (PLS-DA), clustering analysis, and enrichment analysis to identify potential biomarkers of tibolone treatment when the cells were injured with PA. For this, the data was grouped according to the treatments: (i) Tibolone and (ii) PA and PA + tibolone treatments.

This classification revealed a set of metabolites, primarily glycerophospholipids, with significant differences in a relative abundance value recorded for cells treated with tibolone + PA and those treated only with PA. These include phosphatidylcholines (phospholipids) such as the metabolite PC 32: 1, sphingolipids like PI-Cer(d18:0/18:0), and phosphoethanolamines like PE(P-16:0/17:2(9Z,12Z)), which registers representative differences in the endo-metabolome for the PA and PA + tibolone treatments, as well as an increased abundance of the metabolite 16-Glutaryloxy-1alpha, 25-dihydroxyvitamin D3/16-Glutaryloxy-1alpha, 25-dihydroxy-20-epivitamin D3 in the presence of tibolone. These results suggest the importance of tibolone in the generation of neuroprotective metabolites by astrocytes and may be relevant to the development of neurodegenerative processes.

## 2. Results

A total of 3843 features were detected using Charged Surface Hybrid column of a Liquid Chromatography coupled with Electro-Spray Ionization Time of Flight Mass Spectrometry (CSH-ESI-QTOF MS/MS) in endo and exo-metabolome extracts. Using the MS-DIAL program [26]. it was possible to determine the chemical identity of 618 lipids. 249 of the identified compounds corresponded to phosphatidylcholine, 81 compounds were triacylglycerols, 52 lipids were phosphocholine and 43 consisted of to the ceramides. The 31% of the remaining compounds showed a low representation of various types of metabolites (Appendix A). However, 83.9% of the dataset remains unknown.

### 2.1. Applied Treatments Show Different Organization Patterns in PCA and PLS-DA Analysis

To evaluate the data set, four study groups were defined. These partitions were conducted to elucidate the protective effect of tibolone in astrocytes cells exposed to PA. Accordingly, the metabolome distribution of each treatment is showed in the PCA and PLS-DA analysis (Figure 1 and Figure 2). The defined groups were: (i) all the treatments evaluated; (ii) PA + tibolone & PA; (iii) PA + tibolone & tibolone; and (iv) PA + tibolone & tibolone & PA. Likewise, the interaction of each of these groups with the controls was assessed to define the possible interactions among them.

The differences between PCA and PLS-DA analysis founded for all the evaluated conditions are related to mathematical expressions used to calculate the variance into the dataset provided. PCA ignores information regarding the class labels of the samples [27]. The PLS-DA is recognized as a supervised version of a PCA because it achieves a dimensionality reduction keeping the awareness of the class labels. Owing to, PCA represents variances cluster organization of the dataset, but PLS-DA takes known clusters to predict the behavior of the complete dataset. In other words, the distance between the sets changes because the algorithm learns the behavior of the groups that be in the data [27]. According to that, different patterns were described between both analyses (Figure 1).

Nevertheless, PCA and PLS-DA performed with endometabolome data (Figure 1c,d) shows a dramatic change between the position of the control, suggesting when data set conformation have been supervised, the control does not have the same variance as the DMEM and Tibolone data, as it happens with the exometabolome dataset.

The palmitic acid + tibolone (PA + T) & palmitic acid PA treatments comprised a single cluster, suggesting that the application of tibolone does not cause a generalized effect on the abundance of the metabolites evaluated in the presence of PA (Figure 3). However, when evaluating the interaction of the three treatments (PA + T; PA; T) with no controls, it was possible to demonstrate that tibolone is clustered independently from the other group comprising of (PA + T & PA). This corroborates that there is not a global effect on the reported abundances of the metabolites evaluated in the study, subjected to the proposed treatments.

As shown in the dendrograms in Figure 3, the clustering of treatments for the endometabolome (PA + T & T & PA) separated most of the PA-containing data sets from tibolone, which corroborated the effect observed in the PCA and PLS-DA analysis. In the case of the exometabolome, the separation was complete, making the division could provide evidence for the effect of tibolone on the cells studied as distinct from the effect caused by the presence of PA. In addition, it was possible to infer from this behavior that PA + T treatment was not different from PA alone, suggesting that the combination of treatments does not have a differential effect on the exposed cells.

### 2.2. Implementation of Machine Learning Techniques Enhances the Identification of Metabolites Related to Biological Processes

Random Forest analysis expresses the amount of precision that the model obtained when excluding each variable. In other words, the metabolites identified with a high mean decrease value determine the tree’s structure and the exclusion of those metabolites from the model represents a considerable change in the generated model.

A total of 15 metabolites were identified for each group (Figure 4 and Figure 5) and most of them had not been characterized by MS-DIAL [26]. To determine the chemical identity for each metabolite, the CEU mass mediator algorithm was used, but 16% of the metabolites could not be recognized by the algorithm. This program employed databases such as Lipid Maps [28] and KEGG [29] to characterize the metabolites using its mass and reported a list of compounds related to it. In case neither the mass nor any metabolite was reported in these databases, the compound was classified as an unidentified (Appendix A).

Some metabolites displayed similar mass in the databases making the identification process non-specific. For this, as each metabolite had multiple possible identities, we reported every match in Appendix A. Nonetheless, 40% of metabolites were successfully identified and have been reported in Table 1 (endometabolome) and Table 2 (exometabolome).

A total of 33 metabolites of the endometabolome were identified. Of these, 36.3% could only be identified to the amount of the chemical family in which phosphoserines, phosphocholines and phosphoethanolamine were found. The remaining 63.4% were characterized with a singular identity. As for the exometabolomic, 31 metabolites were distinguished. The 64.5% was characterized as a unique compound and the other 35.5% was recognized inside a chemical family, including fatty esters, triacylglycerol’s and glycerophospholipids.

### 2.3. Identified Metabolic Pathways

Enrichment analysis and metabolic pathway mapping was conducted for all metabolites the fully identified in the exo- and endometabolome. However, it was only possible to obtain definitive results for a small subset of metabolites since most of them were not found in the consulted databases (KEGG, SDMP and Lipid Maps). For the endometabolome, metabolic pathway analysis was performed for the metabolite set of the PA + T & PA treatments with MetaboAnalyst (Figure 6, Appendix A). Through this process, 2 possible compounds with a mass of 800.6167 m/z were determined. The first metabolite (C00157) was identified as a phosphatidylcholine associated with 7 possible metabolic pathways: Glycerophospholipid metabolism, Arachidonic acid metabolism, Linoleic acid metabolism, alpha-Linolenic acid metabolism, Biosynthesis of secondary metabolites, Retrograde endocannabinoid signaling, and Choline metabolism in cancer. The second one (C00350), was identified as Phosphatidylethanolamine, which was related to the metabolic pathways of Glycosylphosphatidylinositol (GPI) -anchor biosynthesis and Glycerophospholipid metabolism. Conversely, for the PA + T & T & PA group, 2 metabolites were found with masses of 754.5376 m/z and 836.618 m/z respectively (both with the identifier code C00157). According to the enrichment analysis, these metabolites corresponded to the Phosphatidylcholines or Lecithins. Lastly, the complete set of metabolites with no control, the L-Palmitoylcarnitine metabolite was related to the metabolic pathway of Fatty acid degradation.

For the exometabolome, the Dihomo-gamma-linolenate metabolite (C03242), related to the metabolic pathways of Linoleic acid metabolism and biosynthesis of unsaturated fatty acids, was identified for the PA + T & PA & T treatment group (Figure 6, Appendix A).

The metabolism related with essential acids production and lipids biosynthesis are clearly highlight into the exo- and endometabolome lipidomic analysis. However, most of the compounds identified with a complete chemical identity were inside the cells. That evidence the necessity to try on the metabolite description that could support and extended the chemical knowledge about metabolomic.

## 3. Discussion

In the present study, we conducted the first lipidome-wide investigation of the effects of saturated fatty acid and tibolone treatments on a human astrocytic cell line. It covered 3843 lipid metabolites both in exo- and endometabolome extracts of astrocytic cells stimulated with PA, T, PA + T, DMEM and controls. Previous studies have shown that saturated fatty acid lipotoxicity by metabolites like PA may contribute to NDs including, AD or PD [30] with significant up-regulation of cholesterogenic genes and oxidative metabolism. Moreover, tibolone has been proven to have important effects in brain protection against oxidative damage [22,24].

An extensive separation between the exo- and endometabolome composition of metabolites was performed with PCA and PLSDA (Figure 1 and Figure 2). Importantly, 83.9% of the lipidomic data set in the present study were not identified. This result is related to the lack of robust databases that can be used to determine the compound. It demonstrates the need for further studies focused on precisely defining those metabolites which have not been addressed so far.

Further analyses with the PCA, PLSDA and Random Forest approaches, revealed a greater separation into 4 groups. The defined groups were: (i) all the treatments evaluated; (ii) PA + tibolone & PA; (iii) PA + tibolone & tibolone; and (iv) PA + tibolone & tibolone & PA. In this sense, it was found both by PCA and PLSDA that the PA + T & PA treatment, comprised a single cluster, suggesting that the application of tibolone does not cause a generalized effect on the abundance of the metabolites evaluated in the presence of PA (Figure 3).

Based on dendrograms analysis (Figure 3) it is possible to state that such division can ensure the effect of tibolone on the cells studied is different from the effect caused by the presence of palmitic acid. Likewise, from this behavior, it is possible to infer that treatment of tibolone and palmitic acid doesn’t show a different behavior other than the application of palmitic acid alone, suggesting that the combination of treatments does not have a differential effect on the exposed cells. In other words, a global neuroprotective effect was not evidence with tibolone, nevertheless, the differences could be observed on specific metabolites mentioned below.

Among the different metabolites present in the groups analyzed, there are some important differences between the exo and endometabolome in the treatments. The endometabolome under PA stimulation is characterized by several types of phosphoethanolamines, phosphoserines phosphocholines and glycerophosphocholines. Previous studies have shown that during the development of neurodegeneration, there is a maintained calcium influx and overload in the brain which may increase the breakdown of membrane phospholipides by phospholiphase A2, leading to glycerophosphocholine (GPCh), phosphocholine (PCh), and free choline [31,32]. Other molecules detected in this analysis are sphingolipids, a class of highly enriched lipids in the CNS, which show great diversity and complexity. These molecules are involved in the development and function of the CNS and alterations in its metabolism have been described in multiple NDs, including PD and multiple sclerosis (MS) [33]. Defects in sphingolipid metabolism have been linked to numerous neurological diseases, including Parkinson’s disease and multiple sclerosis. In this aspect, decreased sphingomyelin levels in AD were shown to lead to an increases of ceramide concentration which results in the release of cytochrome C and other related apoptotic proteins [34]. Additionally, enzyme activity and levels of sphingolipid metabolites are typically modulated during pathophysiological conditions. Thus, plasma concentrations may serve as biomarkers for various diseases [33].

On the other hand, in presence of tibolone, metabolites such as 16-Glutaryloxy-1alpha,25-dihydroxyvitamin D3/16-Glutaryloxy-1alpha,25-dihydroxy-20-epivitamin D3, which are derivatives of vitamin D3 [35], were promoted as suggested by Random Forest analysis (Table 2). Previous studies have shown that vitamin D3 is essential for the brain metabolism and homeostasis, demonstrating receptors in all brain cell types [36]. For instance, it has been proven that Vitamin D3 reduces the amyloid-β accumulation and improves cognition in animal models due to its anti-inflammatory and antioxidant properties. Moreover, a recent study in rats evidenced that lipopolysaccharide stimulation in astrocytes enhanced the expression of vitamin D receptor and the D3 converting enzyme Cyp27B1, leading to the suppression of the expression of proinflammatory cytokines such as tumor necrosis factor-α (TNF-α), interleukin-1β (IL-1β) and vascular endothelial growth factor (VEGF) as well as decreased astrocytic activation (Jiao et al. 2017). Lastly, vitamin D3 deficits has been related with a greater risk of dementia and cognitive impairment in human patients [37,38,39].

It was also found that for the PA + T & PA treatments, the tested metabolites were related to different metabolic pathways including, glycerophospholipid metabolism, Arachidonic acid metabolism, Linoleic acid metabolism and alpha-Linolenic acid metabolism (Figure 6). Importantly, these metabolic pathways are associated with neuroinflammatory and myelination processes [40,41]. In this sense, decreased levels in both linoleic acid (LA) and α-linolenic acid, have been implicated to an increased vulnerability to AD, especially in brain regions such as the middle frontal and inferior temporal gyri [42]. Moreover, in a recent study, it was shown that the deuterated form of linoleic acid (D4-Lnn) was able to decrease necrosis and apoptotic cell death in brain cortical cells of newborn mice, through the inhibition of calcium ions production and ROS overproduction [17]. Similarly, Arachidonic acid, a polyunsaturated omega-6 fatty acid, has been shown to be increased in primary culture astrocytes following Lipopolysaccharide (LPS) inflammatory stimulation [43]. Although these metabolic pathways were reported during the analysis of our study, further research is needed to properly understand the metabolic pathways that can be affected by PA and tibolone in both the exo and the endometabolome.

Although to our knowledge, this work is the first lipidomic analysis of human astrocytes in lipotoxic conditions, it has some major limitations. First, in vitro cell line models like NHA may not reproduce aspects of astrocytic environment in the brain, which are relevant to its response to PA and tibolone. Second, vast fraction of the identified compounds has not biological information regarding their function in the brain or in human physiology, including molecules present in both exo- and endometabolome components such as: (3beta,24S,24′S)-fucosterol epoxide, 3beta-hydroxy-stigmast-5-en-7-one, (25R)-5alpha,6alpha-epoxy-24R,26R-dimethyl-26,27-cyclo-cholestan-3beta-ol, heptadecynoic acid, among others. However, our data strongly suggest the importance of some lipidic categories affected by the PA insult, such as phosphoethanolamines, phosphoserines phosphocholines and glycerophosphocholines. Third, for a proper understanding of the effects of the studied lipid components in the brain it would be of great importance to develop further studies in cocultured neuron-astrocytes systems, organoids or animal models exposed to PA and tibolone. In this aspect, previous studies have shown that astrocytic lipid droplets have neuroprotective functions against lipotoxicity through mitochondrial β-oxidation in response to neuronal activity creating a detoxification \environment in a Sprague Dawley rat model [44]. Moreover, a recent article showed that Neurotoxic reactive astrocytes are able to induce cell death through the formation of long chain saturated lipids, suggesting a pathogenic mechanism that could be present in neurodegeneration [45].

Finally, is important to mention that various studies have focused on the importance of brain lipidomics in the development of neuronal pathologies such as PD, AD, MS and schizophrenia among others [16,46,47,48]. Such studies have mainly addressed the importance of cell membrane components such as glycherophospholipids, cholesterol, and sphigolipids and their alterations in neurodegenerative contexts.

## 4. Materials and Methods

### 4.1. Cell Cultures

Three different lots (#0000612736, #0000565612, #0000514417) of primary Normal Human Astrocyte (NHA) cells (Lonza, Basel, Switzerland, Catalog CC-2565) from three different donors (2 female, 1 male) were cultured in Astrocyte Basal Medium (Lonza, Basel, Switzerland) supplemented with SingleQuots supplements (Lonza, Basel, Switzerland). The cells (pass 1) were seeded in 6-well plates at a confluence of 10,000 cells/cm^2^ and grown for 12 days in a humidified incubator at 37 °C and 5% CO_2_. This cell line has astrocytic morphology and expresses GFAP [49,50].

### 4.2. Palmitic Acid Treatment

NHA cells were seeded in 48-well plates at 5.000 cells/cm^2^ for 12 days, washed with 10X PBS, and starved in free serum DMEM without L-Glutamine, phenol red and supplements (Lonza) for 6 h. Cells were then treated with free-serum DMEM containing 2 mM PA for 24 h (Sigma, St Louis, MO, USA), 1.35% bovine serum albumin (BSA) (Sigma Sigma, St Louis, MO, USA, lot A2153) as a carrier protein, and 2 mM carnitine (Sigma, St Louis, MO, USA) as a transporter into the mitochondrial matrix. This concentration induced 50% cytotoxicity in a sensitivity experiment comparing 6 concentrations from 100 µM to 2 mM for 24 h. Cells in control condition received serum-free DMEM with the same BSA and carnitine concentrations but no palmitic acid (PA).

### 4.3. Tibolone Pre-Treatment

Cells were pre-treated with tibolone prior to the addition of PA. Tibolone (Lot T0827, Sigma, St Louis, MO, USA) was dissolved in DMSO as a stock solution at 40 mM, and further dilutions were prepared with serum-free DMEM to a final concentration of 0.000025%. Different times and concentrations of tibolone treatment were tested, and 10 nM of tibolone for 24 h was found to best preserve cell viability upon PA treatment.

### 4.4. Metabolite Extraction

For the extraction of total metabolites, we used a modified protocol by [51]. Briefly, the culture medium was discarded and washed 2 times with 1 mL of 1X PBS at 37 °C by pipetting. 1.5 mL of HPLC grade methanol at −80 °C was added. The cells were gently detached with a scraper and the cell contents were transferred to a 2 mL Eppendorf. 10 µL of internal standard (norvaline dissolved in pyridine) were subsequently added and incubated in an ice bath for 10 min with a vortex, and then chilled in liquid nitrogen for 2 times. Extracts were centrifuged at 4 °C, 12.000 rpm for 5 min and the supernatant was transferred to a new tube followed by two extractions with 250 µL of cold methanol at 80%. After that, the supernatants were combined, and the cell residues were discarded. Finally, extracts were dried under nitrogen flow or lyophilized. Each treatment (PA, tibolone, PA + tibolone, control) had 3 biological replicates (i.e., lots from different donors) and 2 technical replicates (i.e., samples from the same lot), for a total of 30 samples.

### 4.5. Derivatization of Samples/Standards

Sample preparation was performed using the protocol of Matyash V. et al. [52]. Briefly, The dried samples are spiked with 975 uL of MeOH:MTBE and QC mix. Then, 188 uL of water (LC grade) is added and shaked at 4 °C and for 6 min. Subsequently the samples were centrifuged during 20 s at 14,000 g both phases were separated and placed in different tubes and dried using a centrivap. The upper phase was reconstituted adding 110 uL MeOH:Tol (9:1) + CUDA (50 ng/mL), vortexed (10 s), sonicated (5 min) and centrifuged (2 min at 16,100 g). Then, 45 uL was placed in an amber vial with microinsert.

### 4.6. Lipidomics Instrumental Analysis

Hydrophilic Interaction Liquid Chromatography Coupled to Electrospray Ionization Quadruple Time-of-Flight Mass Spectrometry (HILIC-ESI QTOF MS/MS) was conducted to identify and quantify the astrocytes metabolites present in baseline control and under PA and Tibolone treatment. Analyses were performed using an Agilent 1290 Infinity LC system (G4220A binary pump, G4226A autosampler, and G1316C Column Thermostat) coupled to a SCIEX Triple TOF mass spectrometer. Polar compounds were separated on an Acquity UPLC BEH Amide Column, 130Å, 1.7 µm, 2.1 mm X 150 mm maintained at 45 °C at a flowrate of 0.4 mL/min. Solvent pre-heating (Agilent G1316) was used. The mobile phases consisted of: Water, 10 mM Ammonium Formate, 0.125% Formic Acid (A) and Acetonitrile: Water (95/5, *v*/*v*), 10 mM Ammonium Formate, 0.125% Formic Acid (B).

The gradient was as follows: 0 min 100% (A); 0–2 min 100% (A); 2–7.7 min 30% (A); 7.7–9.5 min 60% (A); 9.5–10.3 min 70% (A); 10.3–12.8 min 0% (A); 12.8–16.8 min 0% (A. A sample volume of 1 µL for positive mode and 3 µL for negative mode was used for the injection. Sample temperature was maintained at 4 °C in the autosampler.

SCIEX Triple TOF 6600 mass spectrometers were operated with electrospray ionization (ESI) performing full scan in the mass range m/z 50–1200. Number of cycles in MS1 is 1667 with cycle time of 500 ms and accumulation time 475 ms. Mass spectrometer parameters were as follows (positive mode) Gas Temp 300 °C, gas pressures in psi units with: GS1 and GS2 50 psi, CUR: 35. ISVF is 4500 V and DP and CE are 10 V and 100 V. The Chromatographic analyses and MS analyses were conducted at West Coast Metabolomics Center (WCMC) following a modified protocol by [26,52].

### 4.7. Data Processing

The metabolites of the complete profile were identified using the CEU mass mediator algorithm. This program used the molecular weight of the metabolites to find compounds reported in databases that had the same weight. The list was filtered using different resources to obtain a suitable molecule and the astrocyte genome-scale metabolic model as well as Recon 3D model were used to identify the correct molecule. The metabolomic analysis protocol used in this research has been previously described [53].

Several tests were conducted to determine the most relevant metabolites that differentiate each condition evaluated. Initially, the abundance quantification for each metabolite was received as raw data. Normalization was applied to possible biases during the data processing for these unwanted peak intensity differences and to stabilize the variance within the database. There are several ways to normalize the data: sum, mean, or median. First, however, it is essential to choose the proper method according to the data. In this study, we used the normalization by sum, where each value in a row (metabolite abundance per sample) was divided by the total sum of the row and multiplied by 100 [54]. The method was chosen for the number of measures processed and the best quality check using graphic representation. Finally, a Shapiro-Wilk test (normality test) was applied to determine the normality distribution of the data. Thus, a non-parametric analysis was indicated for this purpose.

### 4.8. Data Analysis

Principal component analysis (PCA) is a mathematical approach used to reduce the dimensionality of the data when responding to a multivariate data set [55]. Once the PCAs were performed, it was impossible to identify clear differences between the groups of data evaluated with this technique. Therefore, a cluster and discriminant analysis were performed to differentiate the groups of data considered [53]. This analysis was conducted using the MetaboAnalyst 5.0 platform [56,57]. Clustering analysis was mainly used to organize samples, characterized by a set of variables, into groups. The major purpose was to create clusters with two complementary characteristics: (i) Maximum internal homogeneity (intra-group similarity) and (ii) High external heterogeneity (inter-group differences). These analyzes are divided into hierarchical and non-hierarchical. In both cases, similarity between samples is measured employing Euclidean or Mahalanobis distances [53]. In this study, Euclidean distance was used. The Partial Least Squares Discriminant Analysis (PLS-DA) was used to identify the principal components of the independent variable. PLS-DA is usually considered a supervised version of PCA because it achieves the reduction of dimensionality considering the classification of variables [27] The VIP scores were calculated as a weighted sum of the squared correlations between the PLS-DA components and the original variable [58]. This score was used to identify the most 15 variable metabolites among the evaluated treatments to determine potential biomarkers of the neuroprotective effects of tibolone.

### 4.9. Machine Learning Approach

Random forest has been widely used in microarray and single nucleotide polymorphism due to its simple theory, fast speed, stability, and insensitivity to noise, with little or no overfitting [53]. This technique was used to find the most significant compounds in the samples, thereby meaning these metabolites determine the tree’s structure. Each tree was created using a tree classification algorithm and the most popular class-based was selected on bootstrap sampling to perform inference among the data which is resampled [53]. The default method for measuring the importance of a variable is the Gini score. It measures the increase in prediction error if the values of the variable are permuted across observations [53]. It means this feature importance score provides a relative ranking of the spectral features [59]. Therefore, the variable is more important if the Gini score is higher than the others [53]. Relevant and important metabolites were identified using the Gini score.

### 4.10. Enrichment Analysis

An enrichment analysis was performed using MetaboAnalyst software [57] to identify the metabolite pathways based on Kyoto Encyclopedia of Genes and Genomes (KEGG) database [29]. Additionally, other databases were used including PubChem [60], BioCyc [61] and Human metabolome database [62]. Identifiers were directly related to a metabolic network or a biological context. For this, the metabolites with the highest Variable importance in projection (VIP) were introduced in the pathway analysis package of MetaboAnalyst. Then, the species *Homo sapiens* was selected using the betweenness-centrality algorithm. Finally, a scatter plot was generated to visualize the pathways with the greater impact.

## 5. Conclusions

Our findings characterize the lipidomic-wide effects of PA and tibolone in cultured human astrocytes, underscoring the comprehensive metabolic dysregulation induced by this saturated fatty acid. Furthermore, this work expands our understanding of the cellular mechanisms by which saturated fatty acids may contribute to neurodegenerative changes and suggests the importance of tibolone in the protection of astrocytic metabolism under inflammatory conditions.

## Figures and Tables

**Figure 1 ijms-23-02474-f001:**
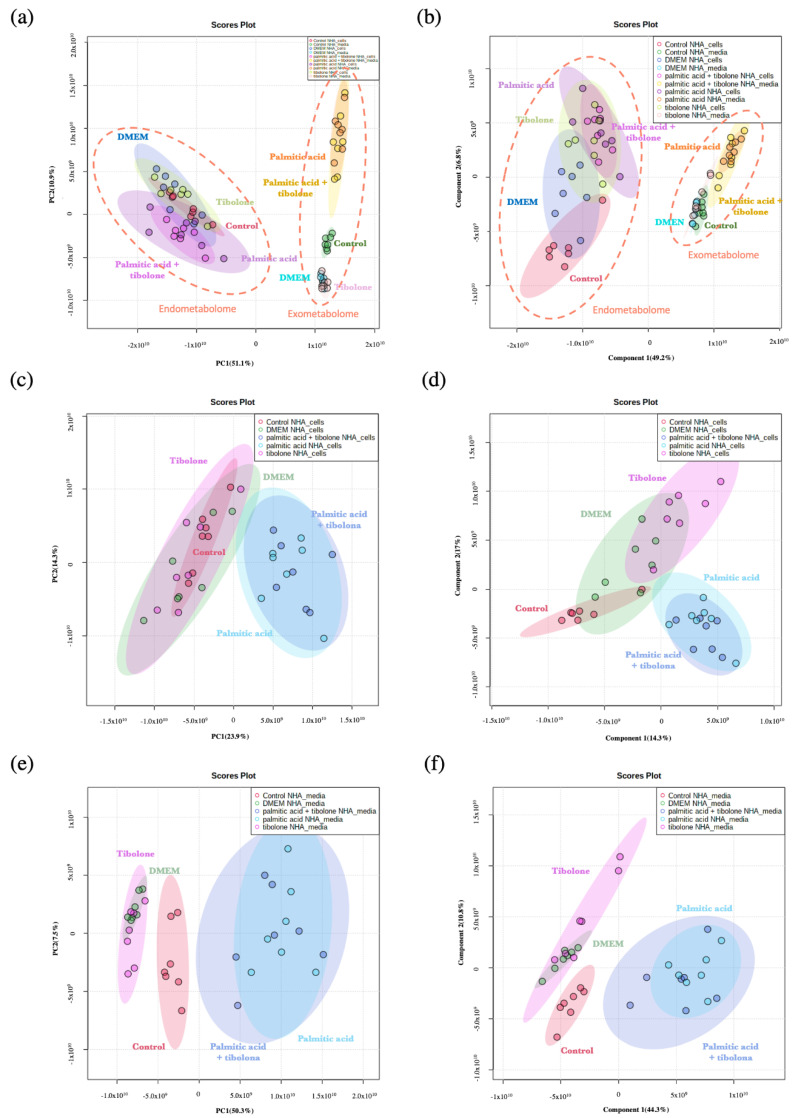
Analysis for the complete data set (**a**) Principal component analysis to complete data for (left) endometabolome, the color patterns correspond to DMEM (blue), palmitic acid + tibolone (magenta), tibolone (light green), control (red), palmitic acid (purple). The exometabolome (right) represent palmitic acid (orange), control (dark green), DMEM (light blue), tibolone (pink) and palmitic acid + tibolone (yellow). (**b**) Partial least squares discriminant analysis for (left) endometabolome and (right) exometabolome. The color patterns are the same as PLS-DA. (**c**,**d**) Specific analysis for the endometabolome (**c**) Principal component analysis and (**d**) partial least squares discriminant analysis, where treatments are represented with tibolone (pink), control (red), DMEM (light green), palmitic acid (light blue), palmitic acid + tibolone (dark blue). (**e**,**f**) Specific analysis for the exometabolome. The pattern colors are the same as the endometabolome analysis.

**Figure 2 ijms-23-02474-f002:**
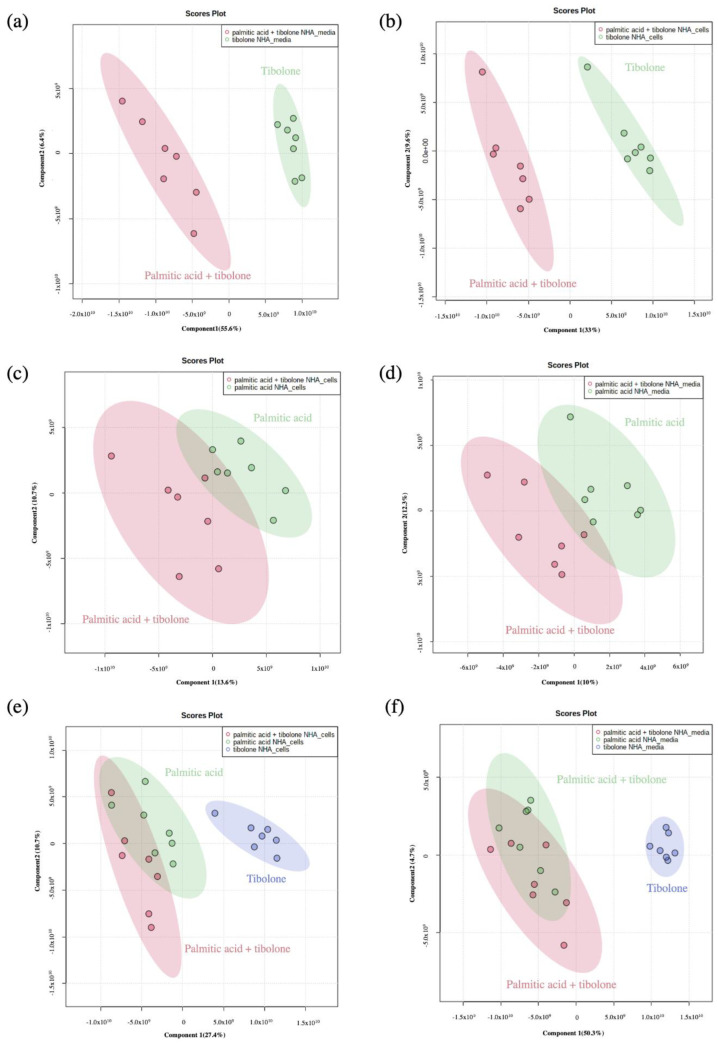
Partial least squares discriminant analysis (PLS-DA). (**a**,**b**) palmitic acid + tibolone (red) and tibolone (green). (**c**,**d**) palmitic acid + tibolone (red) and palmitic acid (green). (**e**,**f**) palmitic acid + tibolone (red) and palmitic acid (green) and tibolone (blue). (**a**,**c**,**e**) endometabolome and (**b**,**d**,**f**) exometabolme. The samples clustering together according with the tibone treatment. It was possible to determine that sampleas are not separate as cluster for PA and PA + T threatments.

**Figure 3 ijms-23-02474-f003:**
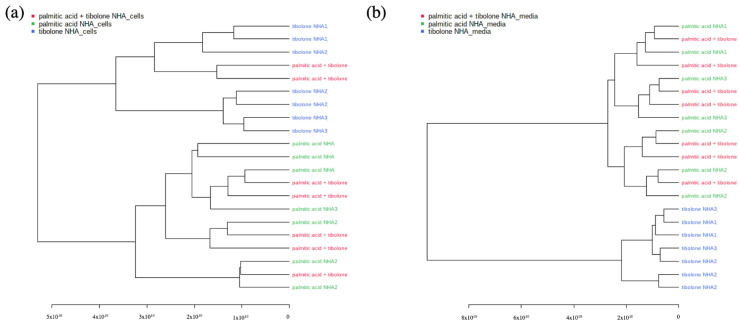
Dendrograms (**a**) (PA & T & PA + T)—Endometabolome (**b**) (PA & T & PA + T)—Exometabolome. The graph shows the complete separation between tibolone and the two remaining treatments (PA + T & PA) for the data corresponding to exometaboloma. However, this behavior is not evidenced in endometabolome, where there are two variables related to the treatments of palmitic acid + tibolone within the tibolone dataset. The dedrograms reflect the PCA clustering organization.

**Figure 4 ijms-23-02474-f004:**
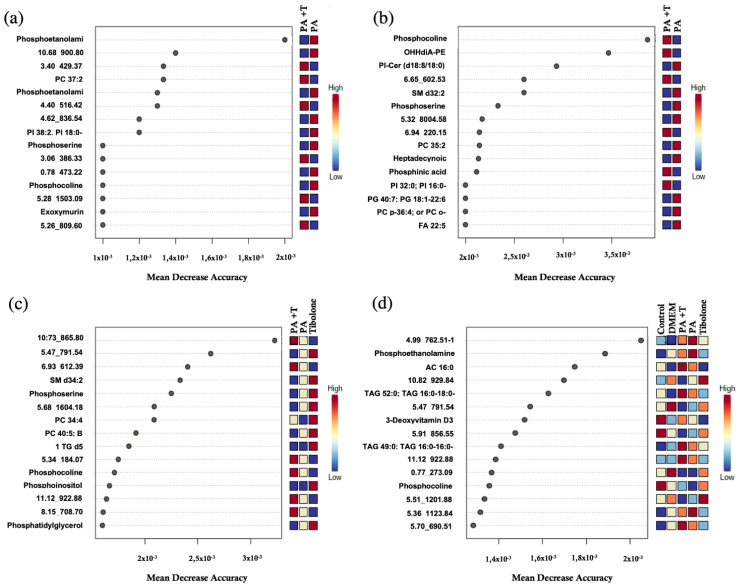
Random Forest for endometabolone analysis. (**a**) PA + T & PA (**b**) PA + T & T (**c**) PA + T & PA & T and (**d**) complete treatments. The list of the first 15 metabolites highlighted by their mean decrease accuracy value is presented for each evaluated treatment set. These metabolites have been reviewed in the literature and related to metabolic processes in humans within the present study.

**Figure 5 ijms-23-02474-f005:**
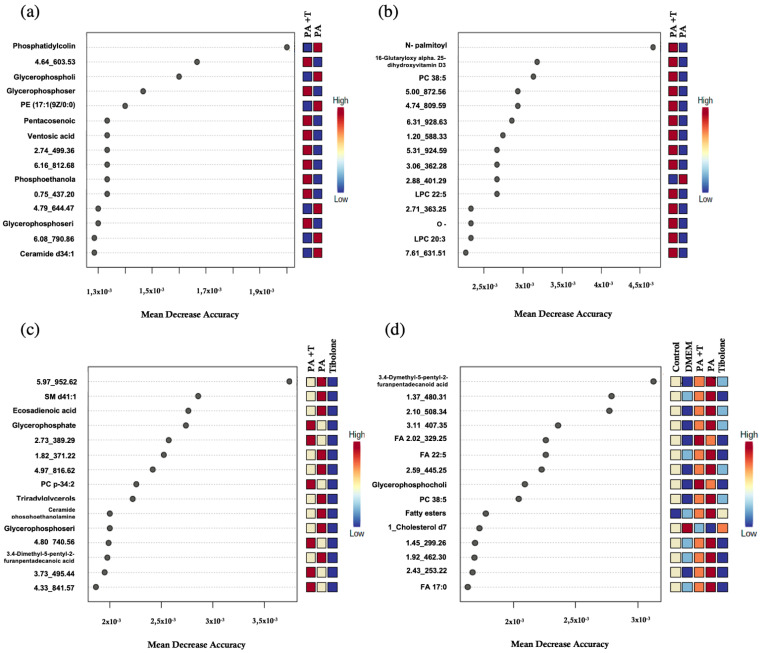
Random Forest for exometabolone analysis. (**a**) PA + T & PA (**b**) PA + T & T (**c**) PA + T & PA & T and (**d**) complete treatments. The list of the first 15 metabolites highlighted by their mean decrease accuracy value is presented for each evaluated treatment set. These metabolites have been reviewed in the literature and related to metabolic processes in humans within the present study.

**Figure 6 ijms-23-02474-f006:**
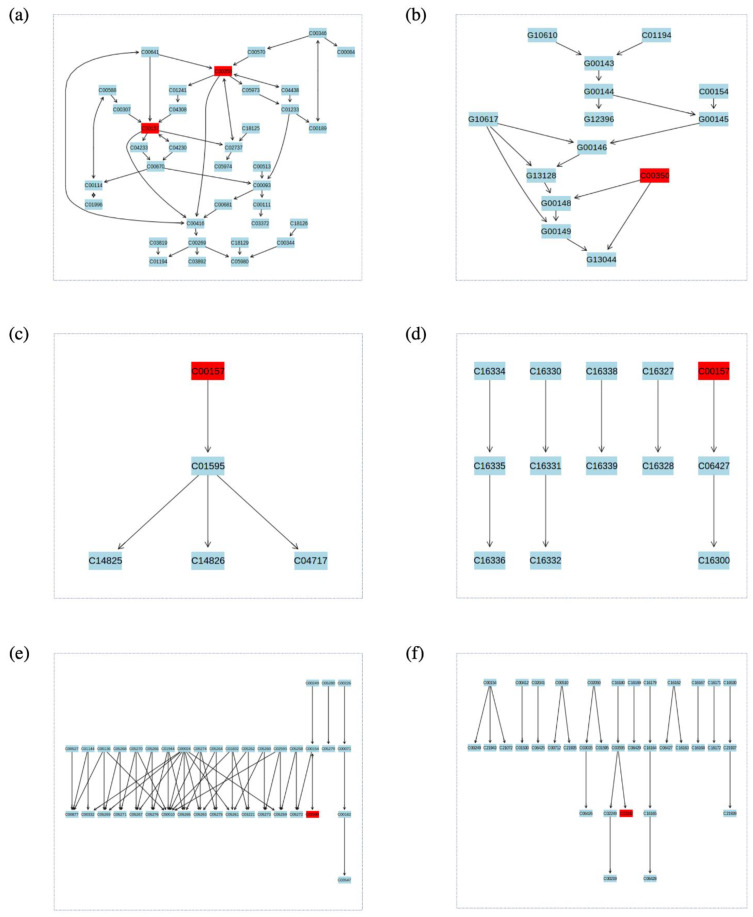
Enrichment analysis performed with identified metabolites for endo- and exometabolome, each one represents a significant metabolic pathway (**a**) Glycerophospholipid metabolism (**b**) Glycosylphosphatidylinositol (GPI) biosynthesis (**c**) Linoleic acid metabolism (**d**) alpha-Linolenic acid metabolism (**e**) Fatty acid degradation (**f**) Biosynthesis of unsaturated fatty acids. In red, metabolites identified with the analysis was highlighted. C00157 is phosphatidylcholine or lecithin; C00350 is (3-Phosphatidyl) ethanolamine; C02990 is L-Palmitoylcarnitine and C03242 is Dihomo-gamma-linolenate.

**Table 1 ijms-23-02474-t001:** List of annotated metabolites using CEU mass mediator for endometabolome. Metabolites written in italics were characterized only to chemical family identification.

Treatments	m/z	Adducts	Formula	Δm/z (ppm)	Compound Type
**PA + T & PA**	*702.5087*	[M-H]-			*Phosphoethanolamine*
	800.6167	[M+H]+	C_45_H_86_NO_8_P	0	PC 37:2;
	*780,5547*	[M-H]-	C44H80NO8P	0	*Phosphoethanolamine*
	889.5812	[M-H]-	C47H87O13P	0	PI 38:2; PI 18:0-20:2;
	*770.5337*	[M-H]-	C42H78NO9P	1	*Phosphoserine*
	*828.5766*	[M-H]-	C45H84NO10P	1	*Phosphocholine*
	531.4769	[M+H]+	C35H62O3	1	Epoxymurin A/ 30-(-2-(O-2-hydroxy-ethane)-3-hydroxy-propane)-hopane
	*524.3719*	[M+H]+	C26H54NO7P	2	*Phosphocholine*
**PA + T & T**	636.3497	[M-H]-			OHHdiA-PE
	810.5923	[M+H]+	C42H84NO11P	8	PI-Cer(d18:0/18:0)/PI-Cer(d20:0/16:0)
	673.526	[M+H]+	C37H73N2O6P	3	SM d32:2;
	*798.5584*	[M+H]+	C44H80NO9P	7	*Phosphoserine*
	830.5904	[M+HAc-H]-			PC 35:2;
	267.2332	[M-H]-	C17H32O2	1	Heptadecynoic acid
	480.309	[M-H]-	C23H48NO7P	1	Phosphinic acid
	809.5197	[M-H]-	C41H79O13P	1	PI 32:0; PI 16:0-16:0;
	819.5187	[M-H]-	C46H77O10P	1	PG 40:7; PG 18:1-22:6;
	824.5814	[M+ HAc-H]-			PC p-36:4; or PC o-36:5;
	329.2489	[M-H]-	C22H34O2	1	FA 22:5;
	759.5662	[M+HAc-H]-			SM d34:2;
**PA + T & PA & T**	*816.5759*	[M+HAc-H]-			*Phosphoserine*
	754.5376	[M+H]+	C42H76NO8P	1	PC 34:4;
	836.618	[M+H]+	C48H86NO8P	2	PC 40:5; B
	*753.5475*	[M+H]+	C43H77O8P	6	*Phosphocholine*
	*797.5145*	[M-H]-	C47H75O8P	2	*Phosphoinositol*
	*771.5181*	[M-H]-	C42H77O10P	0	*Phosphatidylglycerol*
	890.7687/874.7944/869.8343	[M+K]+_[M+Na]+_[M+NH4]+	C54H97D5O6	8	1_TG d5 17:0/17:1/17:0; iSTD
	*756.553*	[M+H]+	C42H78NO8P	1	*Phosphoethanolamine*
**Complete**	400.343	[M+H]+	C23H45NO4	2	AC 16:0;
	885.7905/880.8353	[M+Na]+_[M+NH4]+	C55H106O6	3	TAG 52:0; TAG 16:0-18:0-18:0;
	369.3513	[M+H]+	C27H44	1	3-Deoxyvitamin D3
	838.7822	[M+NH4]+			TAG 49:0; TAG 16:0-16:0-17:0;
	*766.5727*	[M+H]+	C44H80NO7P	2	*Phosphocholine*

**Table 2 ijms-23-02474-t002:** List of annotated metabolites identified with CEU mass mediator for exometabolome. Metabolites written in italics were characterized by chemical family identification.

Treatments	m/z	Adducts	Formula	Δm/z (ppm)	Compound
**PA + T & PA**	836.6165	[M+H]+			*Phosphatidylcholine*
	*835.5348*	[M-H]-			*Glycerophospholipids*
	802.5609	[M-H]-			*Glycerophosphoserines*
	466.2932	[M-H]-	C_22_H_46_NO_7_P	2	PE(17:1(9Z)/0:0)/PC(14:1(9Z)/0:0)
	381.3737	[M-H]-	C_25_H_50_O_2_	0	Pentacosenoic acid/Mycolipenic acid (C25)
	405.3214	[M-H]-	C_22_H_44_O_6_	1	Ventosic acid
	730.57	[M-H]-			*Phosphoethanolamin/Glycerophosphocholines*
	842.5916	[M-H]-			*Glycerophosphoserines*
	572.4818/596.528	[M+Cl]- _[M+HAc-H]-	C_34_H_67_NO_3_	1	Ceramide d34:1;
**PA + T & T**	354.3014	[M-H]-	C_21_H_41_NO_3_	0	N-palmitoyl proline/N-oleoyl alanine
	547.3674	[M+H]+	C_32_H_50_O_7_	8	16-Glutaryloxy-1alpha,25-dihydroxyvitamin D3/16-Glutaryloxy-1alpha,25-dihydroxy-20-epivitamin D3
	866.5909	[M+HAc-H]-	C_48_H_84_NO_10_P	0	PC 38:5; A
	628.3619	[M+HAc-H]-	C_30_H_52_NO_7_P		LPC 22:5;
	398.3272	[M+H]+	C_23_H_43_NO_4_	2	O-palmitoleoylcarnitine/trans-Hexadec-2-enoyl carnitine
	604.3614	[M+HAc-H]-			LPC 20:3;
**PA + T & PA & T**	859.6912	[M+HAc-H]-			SM d41:1;
	307.2637	[M-H]-	C_20_H_36_O_2_	2	FA 20:2; (eicosadienoic acid)
	787.6146	[M+H]+			*Glycerophosphates*
	800.5814	[M+HAc-H]-			PC p-34:2; or PC o-34:3;
	811.6775	[M-H]-			*Triradylglycerols*
	715.5757	[M-H]-			*Ceramide phosphoethanolamines*
	814.5593	[M-H]-			*Glycerophosphoserines*
	407.3524	[M-H]-	C_26_H_48_O_3_	2	3,4-Dimethyl-5-pentyl-2-furanpentadecanoic acid
**Complete**	407.3532	[M-H]-	C_26_H_48_O_3_	2	3,4-Dimethyl-5-pentyl-2-furanpentadecanoic acid
	407.3524	[M-H]-	C_26_H_48_O_3_	2	3,4-Dimethyl-5-pentyl-2-furanpentadecanoic acid
	329.2489	[M-H]-	C_22_H_34_O_2_	1	FA 22:5;
	538.351	[M-H]-			*Glycerophosphocholines/Glycerophosphoserines*
	808.5882	[M+H]+	C_46_H_82_NO_8_P	4	PC 38:5; B
	537.4896	[M-H]-			*Fatty esters*
	376.3969	[M+H-H_2_O]+	C_27_H_39_D_7_O	5	1_Cholesterol d7 iSTD
	269.2489	[M-H]-	C_17_H_34_O_2_	1	FA 17:0; (margaric acid)

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
