# Peer review of "Integrated Metabolomics and Lipidomics Reveal High Accumulation of Glycerophospholipids in Human Astrocytes under the Lipotoxic Effect of Palmitic Acid and Tibolone Protection"

_ijms, 2022, doi:10.3390/ijms23052474_

Round 1
Reviewer 1 Report
Manuscript proposed by Cabezas and co-workers (ijms-1543422) entitled “Integrated Metabolomics and Lipidomics Reveal High Accumulation of Glycerophospholipids in Human Astrocytes under the Lipotoxic Effect of Palmitic Acid and Tibolone Protection” presents lipidomic study of Normal Human Astrocytes cultures exposed to toxic concentrations of palmitic acid and tibolone as the protective agent, to establish and identify the potential metabolites that are modulated under these experimental treatments. In my opinion, presented manuscript is incomplete, needs lots of changes and corrections.
My major comments are presented below.
Major concerns:
- Abstract – clearly present the novelty of the proposed method in such kind of study.
- Discuss, how the obtained results may change the medical treatment.
- Introduction – present paragraph including description of the methods used in such kind of lipidomic analysis
- Introduction – the fragment between lines 59 and 78, in my opinion better fit to the discussion, not to introduction. In introduction section, short information about the obtained results and the role of performed studies on the lipidomic should be presented.
- Result section – what is the total amount of identified metabolites? Once the Authors present 3843, on the other hand 3841 can be found. Additionally Authors used different notation – a comma and a dot in numerical values indicate their different meanings, and in work it is often confused.
- Result section – lack of information about the method used in the study – in materials and methods section some short information can be found. Please provide one or two sentences on the beginning of results section, how the analysis was performed.
- Figure 3 is of low quality
- Table 1 – page 8 – what types of ions represent the presented m/z values? M+H, M-H, others? Present calculated m/z values and identified m/z values for the identified compounds. m/z values are presented incorrectly - a comma and a dot in numerical values indicate their different meanings, and in work it is often confused. The same is in the case of table 2. The same is in the case of supplementary data.
- what was the precision of m/z values determination?
- what was the LOD in the performed studies?
- Materials and method section – page 13 – remove the text presented between the lines 306 and 320
- Materials and method section – page 13, line 326 – correct CO2 (use subscript)
- Materials and method section – page 14, line 370 - 4.6. Lipid analysis - Hydrophilic IInteraction Liquid Chromatography Coupled to Electrospray Ionization Quadruple Time-of-Flight Mass Spectrometry (HILIC-ESI QTOF MS/MS) was conducted to identify and quantify the astrocytes metabolites present in baseline control and under PA treatment. – lack of the HILIC LC-ESI-MS method parameters – intstead of this the GC parameters are presented. In the description of HILIC method the second extraction procedure is presented. Which one is correct? This from the par. 4.4 or 4.6?
- page 15, lines 385-387 - The LC/QTOFMS analyses were performed using an Agilent 6890 brand gases, coupled to a LECO Pegasus IV TOF brand mass spectrometer with a Gerstel MPS2 automatic coating change arm – these are not the LC/QTOFMS method parameters? What was the injection volume in the LC-MS analysis> What was the HILIC method gradient? Which column was used? What about the separation methods? What about of ion detection – did the Authors analyzed positive or negative ions? Describe the MS method parameters (m/z range, temperatures in the ion source, nebulizing gas flow, others).
- line 375, page 14 correct uL
- page 15, line 394 - Mass spectrum was acquired from 85 to 500 m/z at – it is wrong based on the data presented in the Table 1 and 2. There are m/z values higher that 500 m/z.
- Materials and method section – lack of MS parameters (temperatures, potentials, collision energies, parent ions, method optimization etc.)
- Materials and method section – determine the HILIC parameters, eluents, temperature, gradient conditions, purity of used solvents, columns – was the method of compound separation optimized?
- Lack of information about the used chemicals (purity, suppliers, amount of used samples)
- how sensitive was the analysis?
- how many times was one sample analyzed?
Check and correct the reference style according to the journal guide
Make changes in the text.
Check and correct English
Author Response
Manuscript proposed by Cabezas and co-workers (ijms-1543422) entitled “Integrated Metabolomics and Lipidomics Reveal High Accumulation of Glycerophospholipids in Human Astrocytes under the Lipotoxic Effect of Palmitic Acid and Tibolone Protection” presents lipidomic study of Normal Human Astrocytes cultures exposed to toxic concentrations of palmitic acid and tibolone as the protective agent, to establish and identify the potential metabolites that are modulated under these experimental treatments. In my opinion, presented manuscript is incomplete, needs lots of changes and corrections.
My major comments are presented below.
Major concerns:
- Abstract – clearly present the novelty of the proposed method in such kind of study.
The abstract was improved to show the novelty of the study and the methods used. Moreover the methodology section was clarified.
- Discuss, how the obtained results may change the medical treatment.
In the newest version of the article, this issue was discussed (lines 364-370).
- Introduction – present paragraph including description of the methods used in such kind of lipidomic analysis
In the introduction section an entire paragraph (lines 59 to 68) was written, explaining the mathematical and computational methods used for the presented analysis:
- Introduction – the fragment between lines 59 and 78, in my opinion better fit to the discussion, not to introduction. In introduction section, short information about the obtained results and the role of performed studies on the lipidomic should be presented.
We considered that this paragraph is fundamental to explain the proposed methodology of the present article so it was maintained in the introduction.
- Result section – what is the total amount of identified metabolites? Once the Authors present 3843, on the other hand 3841 can be found. Additionally, Authors used different notation – a comma and a dot in numerical values indicate their different meanings, and in work it is often confused.
The total amount of signals detected were 3843. From these signals only 618 were annotated and 64 were associated with a chemical family. This results are explained from line 99 to line 103.
- Result section – lack of information about the method used in the study – in materials and methods section some short information can be found. Please provide one or two sentences on the beginning of results section, how the analysis was performed.
This issue was rectified in the newest version specially in lines 451 to 515
- Figure 3 is of low quality
All figures were improved in the newest version of the article.
- Table 1 – page 8 – what types of ions represent the presented m/z values? M+H, M-H, others? Present calculated m/z values and identified m/z values for the identified compounds. m/z values are presented incorrectly - a comma and a dot in numerical values indicate their different meanings, and in work it is often confused. The same is in the case of table 2. The same is in the case of supplementary data.
An extra-column was added in tables 1 and 2 with the information of the adduct and the m/z error in ppm. Commas were replaced.
- what was the precision of m/z values determination?
For the data annotation in MS/MS spectra we use a threshold of 10 ppm (0.001 Da)
- what was the LOD in the performed studies?
In this study the West Coast Metabolomics Center (Fiehn Laboratory) uses a pool of lipids (listed below) to determine the performance of the method. The LOD of these lipids are reported elsewhere (Tsugawa et al, Nature Methods 2015;12:523). Moreover, the data processing method includes a blank subtraction and cleanup of data using MS-FLO (DeFelice et al, Analytical Chemistry 2017; 89:3250). However, the intensity of the noise used for the data processing was 10000 UI, based on the blank response.
|
Identifier |
Annotation |
Species |
InChI Key |
m/z |
ESI mode |
RT |
|
11.62_1076.02_11.62_729.66_11.62_724.70 |
1_CE 22:1; iSTD |
[M+Chol-head-H2O+H]+_[M+Na]+_[M+NH4]+ |
SQHUGNAFKZZXOT-JWTURFAQSA-N |
1076.0156_729.6564_724.6988 |
ESI (+) |
11,62 |
|
5.82_552.54_5.82_534.53_5.83_574.52 |
1_Cer d18:1/17:0; iSTD |
[M+H]+_[M+H-H2O]+_[M+Na]+ |
ICWGMOFDULMCFL-QKSCFGQVSA-N |
552.5377_534.5278_574.5192 |
ESI (+) |
5,82 |
|
4.70_376.40 |
1_Cholesterol d7 iSTD |
[M+H-H2O]+ |
HVYWMOMLDIMFJA-IFAPJKRJSA-N |
376,3969 |
ESI (+) |
4,70 |
|
0.72_341.28 |
1_CUDA iSTD |
[M+H]+ |
HPTJABJPZMULFH-UHFFFAOYSA-N |
341,2807 |
ESI (+) |
0,72 |
|
4.16_495.34_4.16_479.37_4.16_474.42 |
1_DG 12:0/12:0/0:0; iSTD |
[M+K]+_[M+Na]+_[M+NH4]+ |
OQQOAWVKVDAJOI-VWLOTQADSA-N |
495.3401_479.3737_474.4162 |
ESI (+) |
4,16 |
|
3.08_437.27_3.08_421.29_3.09_416.34 |
1_DG 18:1/2:0/0:0; iSTD |
[M+K]+_[M+Na]+_[M+NH4]+ |
PWTCCMJTPHCGMS-YRBAHSOBSA-N |
437.2675_421.2932_416.3392 |
ESI (+) |
3,08 |
|
1.75_510.36 |
1_LPC 17:0; iSTD |
[M+H]+ |
SRRQPVVYXBTRQK-XMMPIXPASA-N |
510,3569 |
ESI (+) |
1,75 |
|
1.27_466.29 |
1_LPE 17:1; iSTD |
[M+H]+ |
LNJNONCNASQZOB-HEDKFQSOSA-N |
466,2946 |
ESI (+) |
1,27 |
|
2.94_362.33_2.94_345.30_2.94_367.28 |
1_MG 17:0/0:0/0:0; iSTD |
[M+NH4]+_[M+H]+_[M+Na]+ |
SVUQHVRAGMNPLW-UHFFFAOYSA-N |
362.3267_345.3005_367.2825 |
ESI (+) |
2,94 |
|
3.46_636.46 |
1_PC 12:0/13:0; iSTD |
[M+H]+ |
FCTBVSCBBWKZML-WJOKGBTCSA-N |
636,4606 |
ESI (+) |
3,46 |
|
6.17_720.56 |
1_PE 17:0/17:0; iSTD |
[M+H]+ |
YSFFAUPDXKTJMR-DIPNUNPCSA-N |
720,5558 |
ESI (+) |
6,17 |
|
4.96_717.59 |
1_SM d18:1/17:0; iSTD |
[M+H]+ |
YMQZQHIESOAPQH-JXGHDCMNSA-N |
717,5923 |
ESI (+) |
4,97 |
|
1.08_286.27 |
1_Sphingosine d17:1; iSTD |
[M+H]+ |
RBEJCQPPFCKTRZ-LHMZYYNSSA-N |
286,2744 |
ESI (+) |
1,08 |
|
9.34_792.65_9.34_776.68_9.34_771.72 |
1_TG (14:0/16:1/14:0)-d5 iSTD |
[M+K]+_[M+Na]+_[M+NH4]+ |
|
792.6533_776.6797_771.723 |
ESI (+) |
9,34 |
|
10.92_890.77_10.92_874.79_10.92_869.83 |
1_TG d5 17:0/17:1/17:0; iSTD |
[M+K]+_[M+Na]+_[M+NH4]+ |
OWYYELCHNALRQZ-ADIIQMQPSA-N |
890.7687_874.7944_869.8343 |
ESI (+) |
10,92 |
|
6.37_546.54 |
1_5-PAHSA-d9; iSTD |
[M-H]- |
|
546,5446 |
ESI (-) |
6,37 |
|
6.07_586.50_6.07_610.54 |
1_Ceramide d18:1/17:0; iSTD |
[M+Cl]- _[M+HAc-H]- |
ICWGMOFDULMCFL-QKSCFGQVSA-N |
586.4976_610.5426 |
ESI (-) |
6,07 |
|
0.58_339.27 |
1_CUDA iSTD |
[M-H]- |
HPTJABJPZMULFH-UHFFFAOYSA-N |
339,2655 |
ESI (-) |
0,59 |
|
1.86_568.36 |
1_LPC 17:0; iSTD |
[M+HAc-H]- |
SRRQPVVYXBTRQK-XMMPIXPASA-N |
568,3614 |
ESI (-) |
1,87 |
|
1.34_464.28 |
1_LPE 17:1; iSTD |
[M-H]- |
LNJNONCNASQZOB-HEDKFQSOSA-N |
464,279 |
ESI (-) |
1,34 |
|
3.07_403.31 |
1_MAG 17:0/0:0/0:0; iSTD |
[M+HAc-H]- |
SVUQHVRAGMNPLW-UHFFFAOYSA-N |
403,3073 |
ESI (-) |
3,07 |
|
3.55_694.47 |
1_PC 12:0/13:0; iSTD |
[M+HAc-H]- |
FCTBVSCBBWKZML-WJOKGBTCSA-N |
694,4661 |
ESI (-) |
3,55 |
|
6.38_718.54 |
1_PE 17:0/17:0; iSTD |
[M-H]- |
YSFFAUPDXKTJMR-DIPNUNPCSA-N |
718,5395 |
ESI (-) |
6,38 |
|
5.19_749.54 |
1_PG 17:0/17:0; iSTD |
[M-H]- |
ZBVHXVKEMAIWQQ-QPPIDDCLSA-N |
749,5355 |
ESI (-) |
5,19 |
|
4.28_828.56 |
1_PI (15:0-18:1)-d7; iSTD |
[M-H]- |
|
828,5635 |
ESI (-) |
4,28 |
|
5.18_775.60 |
1_SM d18:1/17:0; iSTD |
[M+HAc-H]- |
YMQZQHIESOAPQH-JXGHDCMNSA-N |
775,5977 |
ESI (-) |
5,18 |
- Materials and method section – page 13 – remove the text presented between the lines 306 and 320
These lines were removed in the newest version.
- Materials and method section – page 13, line 326 – correct CO2 (use subscript)
This issue was corrected.
- Materials and method section – page 14, line 370 - 4.6. Lipid analysis - Hydrophilic IInteraction Liquid Chromatography Coupled to Electrospray Ionization Quadruple Time-of-Flight Mass Spectrometry (HILIC-ESI QTOF MS/MS) was conducted to identify and quantify the astrocytes metabolites present in baseline control and under PA treatment. – lack of the HILIC LC-ESI-MS method parameters – intstead of this the GC parameters are presented. In the description of HILIC method the second extraction procedure is presented. Which one is correct? This from the par. 4.4 or 4.6?
Although in the whole project we performed three different analyses, in this paper we are reporting just the results of LC Q-TOF MS/MS method. The methodology was corrected and presented in lines 432-452.
- page 15, lines 385-387 - The LC/QTOFMS analyses were performed using an Agilent 6890 brand gases, coupled to a LECO Pegasus IV TOF brand mass spectrometer with a Gerstel MPS2 automatic coating change arm – these are not the LC/QTOFMS method parameters? What was the injection volume in the LC-MS analysis> What was the HILIC method gradient? Which column was used? What about the separation methods? What about of ion detection – did the Authors analyzed positive or negative ions? Describe the MS method parameters (m/z range, temperatures in the ion source, nebulizing gas flow, others).
The complete method for the extraction and processing of the samples was fully described in the new section of the methodology. Lines: 409-422.
- line 375, page 14 correct uL
This issue was corrected.
- page 15, line 394 - Mass spectrum was acquired from 85 to 500 m/z at – it is wrong based on the data presented in the Table 1 and 2. There are m/z values higher that 500 m/z.
Corrected in section 4.6 Lipidomics instrumental analysis (Line 428-449)
- Materials and method section – lack of MS parameters (temperatures, potentials, collision energies, parent ions, method optimization etc.)
Corrected in section 4.6 Lipidomics instrumental analysis (lines 428-449)
- Materials and method section – determine the HILIC parameters, eluents, temperature, gradient conditions, purity of used solvents, columns – was the method of compound separation optimized?
Although in the whole project was performed three different analyses, in this paper we are reporting just the results of LC Q-TOF MS/MS method. The methodology was corrected and presented in lines 428-450
- Lack of information about the used chemicals (purity, suppliers, amount of used samples)
- how sensitive was the analysis?
In this study the West Coast Metabolomics Center (Fiehn Laboratory) uses a pool of lipids (listed below) to determine the performance of the method. All the merit features (LOD, LOQ, Sensitivity, Reproducibility, Precision, Accuracy, Robustness) of the method are reported elsewhere (Tsugawa et al, Nature Methods 2015;12:523). Moreover, the data processing method is also reported (DeFelice et al, Analytical Chemistry 2017; 89:3250).
- how many times was one sample analyzed?
Each treatment (PA, tibolone, PA + tibolone, control) had 3 biological replicates (i.e., lots from different donors) and 2 technical replicates (i.e., samples from the same lot), for a total of 6 samples for each treatment and a total of 30 samples.
Check and correct the reference style according to the journal guide
Make changes in the text.
Check and correct English
All these issues were corrected in the newest version of the article.

Reviewer 2 Report
The paper makes a very good attempt to unravel and understand the lipidomic profile of human astrocytes under the toxic effects exerted by saturated fatty acid palmitic acid. The experimental design is very sound and data analysis has been rigorously conducted. The paper is interesting in its approach to decipher the differences in the lipidomic profile in presence of palmitic acid insult and steroid tibolone protectant. It highlighted the significant lack of robust lipidomic databases. The authors have tried to link their results to the well studies neurodegenerative diseases.
I would recommend the paper for publication with a few minor revisions:
- Font is inconsistent in certain parts of the paper such as discussion.
- Has any such lipidomic profile studies been done in primary neurons. Is it relatable to the present study on the astrocytes. How does the lipidomic profile of astrocytes affect the neurons. Please discuss briefly in the discussion.
- The exo-metabolome profile has the potential to be further explored in biomarker studies. Please mention briefly if any of the metabolites has already been studied as biomarker.
Author Response
The paper makes a very good attempt to unravel and understand the lipidomic profile of human astrocytes under the toxic effects exerted by saturated fatty acid palmitic acid. The experimental design is very sound and data analysis has been rigorously conducted. The paper is interesting in its approach to decipher the differences in the lipidomic profile in presence of palmitic acid insult and steroid tibolone protectant. It highlighted the significant lack of robust lipidomic databases. The authors have tried to link their results to the well studies neurodegenerative diseases.
I would recommend the paper for publication with a few minor revisions:
- Font is inconsistent in certain parts of the paper such as discussion.
Font inconsistencies were corrected in the newest version of the article.
- Has any such lipidomic profile studies been done in primary neurons. Is it relatable to the present study on the astrocytes. How does the lipidomic profile of astrocytes affect the neurons. Please discuss briefly in the discussion.
In the newest version of the article, this was discussed specially in lines 357-373.
- The exo-metabolome profile has the potential to be further explored in biomarker studies. Please mention briefly if any of the metabolites has already been studied as biomarker.
To our knowledge none of the discovered metabolytes has been studied as a biomarker in brain metabolic diseases. However, in the present article is discussed how previous studies have shown the importance of the studied compounds such as phosphoserines phosphocholines and glycerophosphocholines in the context of neurodegenerative diseases (Alzheimer, Parkinson, etc). Lines: 303-345.

Reviewer 3 Report
The manuscript is interesting. But it requires a serious fix.
- Abstract should be more informative. Including more clearly indicate the results obtained.
- The various signaling cascades activated in astrocytes and as a result of neuron-glial interactions should be described in more detail in the introduction. Including the role of Ca2 + ions as a secondary messenger. https://pubmed.ncbi.nlm.nih.gov/34445509/ https://www.researchgate.net/publication/337839964_A_Complex_Neuroprotective_Effect_of_Alpha-2-Adrenergic_Receptor_Agonists_in_a_Model_of_Cerebral_Ischemia-Reoxygenation_In_Vitro
- Figure legends should be enlarged and their quality improved. Figure 1a is very difficult to read
- Figure 5. The captions in the figure have moved. The meaning of the picture is difficult to understand.
- Figure 6 is of great interest, but a more detailed description of it in the manuscript results is needed. Ideally, the section “Identified metabolic pathways” should have a small clear conclusion.
- Cell culture images are essential. Including confirmation that these are astrocytes, for example, staining with GFAP. The cultivation conditions should be described in more detail, including from which cell passage the work began, the density of the culture during inoculation, and on what day the confluence was achieved.
- The discussion should discuss the effects of deuteration of fatty acids as cytoprotectants. https://pubmed.ncbi.nlm.nih.gov/34948013/ Including the possibility of astrocyte reactivation under various influences should be discussed. https://pubmed.ncbi.nlm.nih.gov/34884629/
- Statistical methods of analysis should be indicated in the legend below each figure. Whether differences between experimental points are significant should also be clearly indicated with standard error or standard deviation. How many repetitions were for each experiment. N is the number of cell passages, n is the number of cells in one experiment.
Author Response
The manuscript is interesting. But it requires a serious fix.
- Abstract should be more informative. Including more clearly indicate the results obtained.
The abstract was improved in order to be more informative.
- The various signaling cascades activated in astrocytes and as a result of neuron-glial interactions should be described in more detail in the introduction. Including the role of Ca2 + ions as a secondary messenger. https://pubmed.ncbi.nlm.nih.gov/34445509/ https://www.researchgate.net/publication/337839964_A_Complex_Neuroprotective_Effect_of_Alpha-2-Adrenergic_Receptor_Agonists_in_a_Model_of_Cerebral_Ischemia-Reoxygenation_In_Vitro
Signalling cascades were explored in the introduction in the newest version of the article, especially in lines 49-62 and 357-366.
- Figure legends should be enlarged and their quality improved. Figure 1a is very difficult to read
All figures, and figure legends were enlarged and improved.
- Figure 5. The captions in the figure have moved. The meaning of the picture is difficult to understand.
Captions in this and other figures were improved.
- Figure 6 is of great interest, but a more detailed description of it in the manuscript results is needed. Ideally, the section “Identified metabolic pathways” should have a small clear conclusion.
This was further discussed in lines 333-348.
- Cell culture images are essential. Including confirmation that these are astrocytes, for example, staining with GFAP. The cultivation conditions should be described in more detail, including from which cell passage the work began, the density of the culture during inoculation, and on what day the confluence was achieved.
Previous studies by our lab and other groups (for example Rieske et al. 2009 and Azizi et al 2013 ), have shown that the NHA have a basal expression of GFAP. Moreover in the newest version of the article the cultivation conditions were expanded. (lines The cells (pass 1) were seeded in 6-well plates at a confluence of 10000 cells/cm2 and grown for 12 days in a humidified incubator at 37 ºC and 5% CO2.
Azizi SA, Krynska B. Derivation of neuronal cells from fetal normal human astrocytes (NHA). Methods Mol Biol. 2013;1078:89-96. doi: 10.1007/978-1-62703-640-5_8. PMID: 23975823.
Rieske P, Augelli BJ, Stawski R, Gaughan J, Azizi SA, Krynska B. A population of human brain cells expressing phenotypic markers of more than one lineage can be induced in vitro to differentiate into mesenchymal cells. Exp Cell Res. 2009 Feb 1;315(3):462-73. doi: 10.1016/j.yexcr.2008.11.004. Epub 2008 Nov 20. PMID: 19061885.
- The discussion should discuss the effects of deuteration of fatty acids as cytoprotectants. https://pubmed.ncbi.nlm.nih.gov/34948013/ Including the possibility of astrocyte reactivation under various influences should be discussed. https://pubmed.ncbi.nlm.nih.gov/34884629/
In the newest version of the article, a small paragraph was added regarding the deuteration of fatty acids as cytoprotectants. However, the article by Varlamova et al. wasn’t considered of interest for the focus of our article (lipidomics in the context of tibolone and PA stimulation). On the other hand we added a consideration of astrocytic activation in the context of lipid toxicity (lines 327-330).
- Statistical methods of analysis should be indicated in the legend below each figure. Whether differences between experimental points are significant should also be clearly indicated with standard error or standard deviation. How many repetitions were for each experiment. N is the number of cell passages, n is the number of cells in one experiment.
The statistical methods were explained in the methods section, however, for PCA and PLS-DA the quantities used to determine the components were exposed into the figures. On the other hand, statistical analysis does not apply for network predictions. The number of replicates per each experiment was described in the methodology section (cell cultures and metabolites extraction).

Round 2
Reviewer 1 Report
The revised version of the manuscript meets all of my requirements.
Authors made changes according to all of my comemnts and quations.
Manuscript may be accepted for publication.
Reviewer 3 Report
The article has been significantly improved and can be published